# A Novel Underwater Image Enhancement Using Optimal Composite Backbone Network

**DOI:** 10.3390/biomimetics8030275

**Published:** 2023-06-27

**Authors:** Yuhan Chen, Qingfeng Li, Dongxin Lu, Lei Kou, Wende Ke, Yan Bai, Zhen Wang

**Affiliations:** 1Department of Mechanical and Energy Engineering, Southern University of Science and Technology, Shenzhen 518055, China; 12132246@mail.sustech.edu.cn (Y.C.); 11930359@mail.sustech.edu.cn (Y.B.); 12032414@mail.sustech.edu.cn (Z.W.); 2Health Management System Engineering Center, School of Public Health, Hangzhou Normal University, Hangzhou 311121, China; 1030779697@qq.com (Q.L.); zn19980@sina.com (D.L.); 3Institute of Oceanographic Instrumentation, Qilu University of Technology (Shandong Academy of Sciences), Qingdao 266075, China; koulei1991@qlu.edu.cn

**Keywords:** underwater image enhancement, deep learning, composite backbone, composite architectures

## Abstract

Continuous exploration of the ocean has made underwater image processing an important research field, and plenty of CNN (convolutional neural network)-based underwater image enhancement methods have emerged over time. However, the feature-learning ability of existing CNN-based underwater image enhancement is limited. The networks were designed to be complicated or embed other algorithms for better results, which cannot simultaneously meet the requirements of suitable underwater image enhancement effects and real-time performance. Although the composite backbone network (CBNet) was introduced in underwater image enhancement, we proposed OECBNet (optimal underwater image-enhancing composite backbone network) to obtain a better enhancement effect and shorten the running time. Herein, a comprehensive study of different composite architectures in an underwater image enhancement network was carried out by comparing the number of backbones, connection strategies, pruning strategies for composite backbones, and auxiliary losses. Then, a CBNet with optimal performance was obtained. Finally, cross-sectional research of the obtained network with the state-of-the-art underwater enhancement network was performed. The experiments showed that our optimized composite backbone network achieved better-enhanced images than those of existing CNN-based methods.

## 1. Introduction

Exploration of the ocean world has attracted more attention in recent years [1]. As a vital part of image processing for revealing and recognition underwater scenes, underwater image enhancement plays an import role in marine resources and marine military fields [2]. Unfortunately, because of differences in illumination conditions in the complicated underwater environments, the captured images are usually of low quality, with characteristics such as having inauthentic color, blurring, and noise, making the enhancement of such images difficult [3]. In order to accomplish this challenging task, many algorithms have been introduced.

In the early stages, the technical breakthroughs in image enhancement often relied on hardware performances [4]. However, advances in image enhancement not only abated the equipment costs but also indicated being more beneficial for accommodating complicated underwater environments. Researchers used mathematical and statistical analyses to modify the channel values of underwater images to obtain images with high visual quality [5]. Sometimes the enhancement effect for underwater images was regarded as an optics problem, and the corresponding algorithms were established on physical models, such as the dark channel prior (DCP) mechanism [6] and the Retinex model [7]. Some researchers regard underwater image enhancement as a bionic problem [8]. More recently, deep learning gradually became the main underwater image enhancement technology due to its strong modeling ability and efficient inference. Many studies have shown that neural networks obtained enhanced images with higher quality compared to images enhanced by traditional methods [9]. The convolutional neural network (CNN) [10] and generative adversarial network (GAN) [11] were proposed for enhancing underwater images. However, these algorithms are unsuitable because they consume time and space, making them inapplicable.

CBNet shows better performance [12] compared to single-chain networks and provides a broad space for improvement in the network architecture [13]. So herein, CBNet was introduced in the enhancement of underwater images. In order to minimize time and space consumption and meet the actual requirement of marine engineering, CNN models without other embedded algorithms were established. To explore the architectural features of CBNet, comprehensive enhancement research using it in different variants was carried out. An **O**ptimal underwater image **E**nhancement **C**omposite **B**ackbone **Net**work (OECBNet) was finally obtained, which we compared with several state-of-the-art CNN-based methods. Both full-reference indexes and non-reference indexes indicate that the enhancement results of our proposed network are better than recent state-of-the-art CNN-based underwater image enhancement methods. Our proposed network also performs well in real-time image enhancement, with one 350 × 350 underwater image taking only 2.3 × 10^−3^ s to enhance.

The strategy of our research is as follows: Section 1 is the introduction to our research. Section 2 describes the related work, including underwater image enhancement methods, underwater image datasets, and CNN backbone networks. Section 3 describes some architectural details of our backbone network. Section 4 makes a comprehensive exploration of the architectural features of CBNet, and the optimal underwater image-enhancing composite backbone network is obtained from this section. Section 5 evaluates our network performance according to cross-sectional research with other methods. Section 6 makes a conclusion of this paper. Contributions in this paper are summarized as shown in the following three points:

We introduced a composite backbone into CNN-based underwater image enhancement. The composite backbone networks consist of several uniform end-to-end CNN backbones in specific connection strategies. Composite backbone networks were applied throughout the enhancement networks;We conducted a comprehensive study about underwater image enhancement using CBNet in different variants by first investigating the impact of backbone number on the enhancement results. Then, the different connection strategies between backbones and some corresponding pruning strategies were introduced in our research. Moreover, the experiments were conducted in auxiliary losses;We proposed an optimal underwater image-enhancing composite backbone network. The optimized composite backbone network consists of two backbones in full-connected composition (FCC), and two stages are pruned in the lead backbone. The network accurately expresses objects as well as background colors and is well adapted to the engineering field.

## 2. Related Work

### 2.1. Deep Learning-Based Underwater Image Enhancement

Underwater image enhancement methods based on deep learning are classified into five classes according to their network architecture [14]. One idea for constructing image enhancement networks is inspired by semantic segmentation, such as U-Net [15]. Yan et al. [16] proposed a cascaded U-Net for image restoration and achieved enhanced results with less training and inference costs. Moreover, Liu et al. [17] proposed a GAN with an encoder–decoder-based generator, which enabled multi-scale feature extraction and fusion. Other kinds of widely used networks are constructed with multiple branches. Xue et al. [18] enhanced images by predicting the coarse result, veil map, and compensation map according to a multi-branch network. Li et al. [19] introduced white balance (WB), gamma correction (GC), and histogram equalization (HE)-enhanced images in multiple-branch enhancement and conducted a gated fusion. Similarly, Wu et al. [20] proposed a GAN for enhancement and introduced WB, HE, and DCP enhanced images for multi-branch training in the generator. Wang et al. [21] introduced HSV color space for image adjustment; on this basis, Chen et al. [22] conducted an enhancement fusion of RGB, HSV, and Lab color spaces. Some enhancement methods embedded several units for better feature learning in the networks. Li et al. [9] constructed an E-unit for underwater enhancement, and Chen et al. [23] applied the E-unit in a multi-branch network. Further, depth maps or transmission maps are integrated into some networks [24]. However, to implement these networks, special hardware such as deep cameras are required. Based on the excellent performance of GAN, many researchers tend to upgrade their methods by improving the GAN-based algorithm [25] by incorporating multiple generators [26] or multiple discriminators [27] to enhance underwater images.

### 2.2. Underwater Dataset

The dataset is a sensitive part when it comes to deep learning. Researchers use different equipment to try and collect images in a variety of marine environments. Liu et al. proposed a real-world underwater image enhancement dataset (RUIE) [28] with a large number of diverse light-scattering images and rich detection targets. Karen et al. created an underwater object tracking dataset (UOT100) [29] that includes 104 underwater videos for object detection. However, the lack of reference images was always a challenge for these datasets to achieve enhancement. To solve the problem, Anwar conducted research on underwater image synthesis and proposed a method for simulating different underwater scenes [9], which can be applied as full-reference enhancement result evaluations. While retaining the real underwater environment, Li et al. proposed an underwater image enhancement benchmark (UIEB) dataset [19] consisting of a reference subset and a challenging subset. The reference dataset included 890 underwater images as well as corresponding reference images, which partially solved the challenge of the lack of reference images and promoted innovations in underwater image enhancement methods.

### 2.3. Backbone Network

The backbone network plays a significant part in CNN models. Earlier networks were often designed to be as deep as possible [30] and connections as dense as possible [31], but their performance generally degenerated with increasing depth. To improve their quality, researchers added branches in different scales into the backbone networks. ResNet, ResNeXt, and Res2Net were proposed in succession [32]. In addition, Liang et al. proposed a recurrent convolutional neural network (RCNN) [33] which recurrently connected each convolution layer, and the contextual information was efficiently integrated. Meanwhile, Zhang et al. [34] proposed a lightweight backbone network called MobileNet, which greatly reduced the time complexity of the backbone network. In order to expand the sensory field of each pixel, many backbone networks embed multi-scale architectures [35], especially feature pyramid blocks [36]. Another inspiration was from U-Net [15], which introduced downsampling and upsampling in backbone networks [37]. Liu et al. proposed a composite backbone network (CBNet) [12], which brought a breakthrough in object detection. Thereafter, they continued to explore the architecture of CBNet and proposed a general framework with novel connection strategies (CBNetV2) [13]. CBNet has now been widely applied to counter the challenges of object detection and semantic segmentation [38]. Zhao et al. improved adjacent higher-level composition in CBNet, achieving the detection and recognition of images in low-quality underwater videos, which brought CBNet into underwater vision [39]. CBNet has been proven to be excellent for feature extraction and is gradually being applied in underwater image enhancement.

## 3. Backbone Network Details

### 3.1. Global Architecture

The global architecture of CBNet in general is shown in Figure 1. K identical backbones in parallel and the Backbone *i* are compositely connected with backbones from Backbone 1 to Backbone *i* − 1. Output images from the lead backbone are sent to output layers, and output images are obtained from the output convolution block. In addition, each stage obtains images of the same height and weight as input images for pixel-level enhancement. The channel numbers of output images from the stages are constantly the same.

### 3.2. Backbone Architecture

The backbone architecture of our proposed network is as shown in Figure 2, which skillfully ensures that output images are the same height and weight as input images without the shape of input images getting limited. The backbone is empirically constructed with two previous stages, three middle stages, and one last stage in series. Each stage contains a convolutional layer followed by a batch norm layer and a ReLU layer. During the previous stages, the convolutional layers are operated with a 7 × 7 convolution kernel and padding = 3, and the large convolution kernels expand the receptive field for each pixel, thus improving the global image enhancement effect. During the middle stages, the convolutional layers are operated with a 5 × 5 convolution kernel and padding = 2, and the introduction of more middle layers for convolution processing improves the learning ability of the network. During the last stages, the convolutional layers are operated with a 3 × 3 convolution kernel and padding = 1, and last stages are placed before output to accurately enhance the pixelwise detail.

### 3.3. Output Convolution Block

Output blocks play an important role in CNN-based underwater image enhancement for information fusion and channel values’ normalization. However, the importance of output blocks was generally ignored in previous underwater image enhancement networks. Our proposed output convolution block incorporated a 3 × 3 convolutional layer with padding = 1, followed by a Sigmoid layer instead of the traditional ReLU layer. Application of the Sigmoid layer ensures that the pixel channel values of output images lie within the standard range of [0, 1].

## 4. Research on CBNet-Based Underwater Image Enhancement

In this section, we explore the architectural features of CBNet. First, we introduce some experiment details. In Section 4.2, we analyze the impact of backbone number on underwater image enhancement. In Section 4.3, we evaluate the underwater image enhancement performance of different variants. In Section 4.4, we implement pruning strategies in some variants. In Section 4.5, we construct auxiliary losses for partial special variants. Finally, we make a summary of composite backbone architecture and choose a network as the optimal composite backbone network for subsequent comparison.

### 4.1. Implementation Details

We applied the UIEB dataset [22] to our study. To construct a training set, we randomly chose 800 underwater images as well as their corresponding reference images from a reference subset. Other images in the UIEB dataset were treated as test images. All images were preprocessed by stretching to 350 × 350, and training set images were cut into 320 × 320 at random. Our loss function was a weighted sum of *L*1 loss and SSIM loss as follows:(1)$$L(I_{P},I_{R})=\omega_{l_{1}}L_{l_{1}}(I_{P},I_{R})+\omega_{SSIM}L_{SSIM}(I_{P},I_{R})$$
where $\omega_{l_{1}}$ and $\omega_{SSIM}$ are weight parameters used to balance two loss components. Weight parameters are selected as $\omega_{l_{1}}$ = 2, $\omega_{SSIM}$ = 1. The two loss components $L_{l_{1}}$ and $L_{SSIM}$ are calculated as follows: (2)$$L_{l_{1}}(I_{P},I_{R})=\frac{1}{C^{I_{R}}H^{I_{R}}W^{I_{R}}}\sum \left|t^{I_{P}}(i)-t^{I_{R}}(i)\right|$$
(3)$$L_{SSIM}(I_{P},I_{R})=1-\frac{1}{C^{I_{R}}H^{I_{R}}W^{I_{R}}}\sum \frac{2\mu_{I_{P}}(i)\mu_{I_{R}}(i)+C_{1}}{\mu_{I_{P}}^{2}(i)+\mu_{I_{R}}^{2}(i)+C_{1}}\times \frac{2\sigma_{I_{P}I_{R}}(i)+C_{2}}{\sigma_{I_{P}}^{2}(i)+\sigma_{I_{R}}^{2}(i)+C_{2}}$$
where *I_P_* represents the predicted image and *I_R_* represents the reference image. $t^{I_{P}}(i)$ and $t^{I_{R}}(i)$ are the corresponding channel values in a certain pixel *i*. $\mu_{I_{P}}(i)$ and $\mu_{I_{R}}(i)$ are average channel values of a 11^2^ square interval with center *i*. $\sigma_{I_{P}}^{}(i)$ and $\sigma_{I_{R}}^{}(i)$ represent corresponding standard deviations. $\sigma_{I_{P}I_{R}}(i)$ corresponds to the covariance, while parameters *C*_1_ = 0.02 and *C*_2_ = 0.03 are selected to stabilize the SSIM function. *C*, *H*, and *W* mean channels, height, and weight, respectively.

We applied ADAM optimizer for network training and initialized the learning rate as 1 × 10^−3^ and decreased it by 1 × 10^−5^ in each epoch. We extracted the trained network models after 100 epochs, which means it takes nearly two hours to train each model. Pytorch as well as Anaconda were jointly used with Nvidia GTX 3080 GPU for programming.

We used the rest of the 90 pairs of images from the UIEB reference subset for network evaluation, and three classical full-reference indications, namely: mean square error (MSE), peak signal-to-noise ratio (PSNR), and structural similarity index (SSIM), were adopted. A lower MSE or higher PSNR score indicated that the pixel values of the predicted results were closer to references. A higher SSIM indicated that the predicted result was more similar to the reference. However, these indications could only evaluate images from a single channel. To make evaluation results more rigorous, we conducted evaluations from both gray-scale images and RGB images as follows:(4)$$E_{RGB}(I,I)=\left(E(I_{R},{I^{\prime }}_{R})+E(I_{G},{I^{\prime }}_{G})+E(I_{B},{I^{\prime }}_{B})\right)/3$$
(5)$$E_{\mathrm{gray}}(I,I^{\prime })=E(RGB2\;gray(I),RGB2\;gray(I^{\prime }))$$
where *E* represents evaluation functions of indications such as MSE, PSNR, or SSIM. *I* is the enhanced image while *I_R_*, *I_G_*, and *I_B_* are corresponding maps in *R*, *G*, and *B* channels; *I′* is the reference image while *I′_R_*, *I′_G_*, and *I′_B_* are corresponding maps in *R*, *G*, and *B* channels.

### 4.2. Backbone Number Analysis

CBNet was a combination of *K* backbones with the same architecture. We define the backbones as Bone 1, Bone 2, Bone 3, …, Bone *K*. For any defined backbone, if *k* = 1, 2, …, *K* − 1, Bone *k* is considered an auxiliary backbone; if *k* = *K*, Bone *k* is the lead backbone. Generally, there are two approaches for information fusion between backbones, that is, residual fusion and concat fusion. Concat fusion is to amalgamate several images with the same shape in the specified dimension, and the channel number of the output image is the sum of input image channel numbers. Residual fusion is a simple image subtraction. The traditional same-level composition (SLC), as shown in Figure 4a, is applied in backbone number analysis; thus, the *l*th stage transformation of concat-fused information and residual-fused information are shown respectively:(6)$$x_{k}^{l}=F_{C}^{l}(Cat(x_{1}^{l-1}+x_{2}^{l-1}+\cdots +x_{k}^{l-1})),\;2\leq l\leq 7,\;2\leq k\leq K$$
(7)$$x_{k}^{l}=F_{R}^{l}(x_{k}^{l-1}-x_{k-1}^{l-1}),\;2\leq l\leq 7,\;2\leq k\leq K$$
where *Cat* represents the concat fusion of images. $F_{C}^{l}$ is the combinational function of concat-fused information in the *l*th stage, $F_{R}^{l}$ is the combinational function of residual-fused information in the *l*th stage, while $x_{k}^{l}$ is the output image from the *l*th stage in the *k*th backbone. When *l* = 7, $F_{C}^{7}$ and $F_{R}^{7}$ indicate the combinational function of the output block, while $x_{k}^{7}$ indicates the output-enhanced image.

We conducted analyses of underwater image enhancement from networks with different backbone numbers. The output channel number from each stage was set as 24 to meet the internal memory demand. Underwater image enhancement analyses were conducted, as shown in Figure 3, details of analyses from different backbone number as shown in Appendix A. According to SSIM evaluations in Figure 3a, networks with concat fusion obtained the best SSIM score when *K* = 3, and networks with residual fusion obtained the best SSIM score when *K* = 2. According to MSE evaluations in Figure 3b, as the number of backbones increased, MSE_gray_ indicators increased significantly, while MSE_RGB_ from concat fusion networks obtained a weak downward trend. In our case, networks with concat fusion obtained better MSE scores than networks with residual fusion. The advantage of networks with concat fusion became evident according to RSNR evaluation in Figure 3c, and the best PSNR scores were obtained when *K* = 3. According to Figure 3d, the average runtime of concat fusion networks increased linearly with the number of backbones. Compared with concat fusion networks, the runtime of residual fusion networks decreased to some extent. According to the above results, increasing the number of backbones can increase the amount of information and thus increase the possibility of obtaining high-quality results. However, with the increase in information, the time complexity will increase, and the interference information will increase, too. What is more, the introduction of too-deep network enhancement will gradually distort output images. Thus, excessively increasing the auxiliary backbone is often counterproductive. Suitable backbone architectures would appear when *K* = 2 or 3.

### 4.3. Connection Strategy Analysis

The analysis in Section 4.2 showed that concat fusion networks achieve relatively suitable underwater image enhancement results when *K* = 2 or 3. To minimize the time and space complexity, we employed CBNet with two concat-fused backbones for further exploration. To further explore the CBNet architecture, we constructed several variants of CBNet in different connection strategies besides the traditional SLC (Figure 4). Details of different composite strategies are explained in Section 4.3.1, Section 4.3.2, Section 4.3.3, Section 4.3.4 and Section 4.3.5. The output channel number from each stage was set as 32.

#### 4.3.1. Adjacent Higher-Level Composition (AHLC)

To obtain stable image enhancement results, higher-level features can be taken to enhance the lower-level features. In the AHLC variant shown in Figure 4b, output information of the adjacent higher-level stage from the auxiliary backbone was sent to the lead backbone. The *l*th stage transformation of AHLC is shown in Equation (8):(8)$$x_{2}^{l}=F_{C}^{l}(Cat(x_{1}^{l},x_{2}^{l-1})),1\leq l\leq 6$$
where *l* = 1, *x*^0^_k_ indicates input image.

#### 4.3.2. Adjacent Lower-Level Composition (ALLC)

Another line for stabilizing image enhancement results is to take lower-level features to enhance the higher-level features, which is just opposite to AHLC. As shown in Figure 4c, ALLC sent the adjacent lower-level stage output information to the lead backbone. The *l*th stage transformation of ALLC is shown in Equation (9):(9)$$x_{2}^{l}=F_{C}^{l}(Cat(x_{1}^{l-2},x_{2}^{l-1})),\;2\leq l\leq 7$$

#### 4.3.3. Dense Same-Level Composition (DSLC)

As shown in Figure 4d, the output information of each stage from the lead backbone was fused with the output information of non-lower-level stages from the auxiliary backbone and sent to the next stage. The architecture of DSLC is similar to that of DenseNet [33]. The lth stage transformation of DSLC is as shown in Equation (10):(10)$$x_{2}^{l}=F_{C}^{l}(Cat(x_{1}^{l-1},x_{1}^{l},\cdots ,x_{1}^{6},x_{2}^{l-1})),\;2\leq l\leq 7$$

#### 4.3.4. Dense Higher-Level Composition (DHLC)

As shown in Figure 4e, the output information of higher-level stages was sent to the lead backbone. The *l*th stage transformation of DHLC is as shown in Equation (11):(11)$$x_{2}^{l}=F_{C}^{l}(Cat(x_{1}^{l},x_{1}^{l+1},\cdots ,x_{1}^{6},x_{2}^{l-1})),\;1\leq l\leq 6$$

#### 4.3.5. Full-Connected Composition (FCC)

To connect the auxiliary backbone to the leading backbone as compactly as possible, each auxiliary stage backbone output information was sent to all lead backbone stages, as shown in Figure 4f. The *l*th stage transformation of FCC is as shown in Equation (12):(12)$$x_{2}^{l}=F_{C}^{l}(Cat(x_{1}^{1},x_{1}^{2},\cdots ,x_{1}^{6},x_{2}^{l-1})),\;1\leq l\leq 7$$

We then evaluated underwater image enhancement results from networks with different composite strategies described above. The output channel number from each stage was set as 32. Evaluations of the enhanced underwater images from different connection strategies are shown in Table 1, details as shown in Appendix A.

According to the evaluations, FCC achieved the best scores in SSIM indications, DHLC achieved the best scores in MSE and PSNR indications, and AHLC achieved the best scores in MSE_gray_ and PSNR_gray_. The results revealed that FCC obtained enhancement images almost similar to reference images, DHLC obtained enhancement images closely resembling reference images, while AHLC obtained enhancement images closer to reference images from gray-scale than other connection strategies.

Further, Table 2 shows the runtime comparison among networks with different connection strategies. According to the results, due to the reduction in the connection between the lead backbone and the auxiliary backbone, ALLC completed underwater image enhancement within the shortest runtime (2.55 × 10^−3^ s per image on average). Meanwhile, DHLC constructed pretty complex connections between the lead backbone and the auxiliary backbone, thus completing underwater image enhancement in the longest runtime. Notably, FCC constructed similar concat connections before each stage, which simplified the network procedure; thus, FCC also enhanced images relatively fast, with an average runtime of only 2.59 × 10^−3^ s.

### 4.4. Pruning Strategy Analysis

To reduce the excessive connection between network backbones, we simplified the architecture of CBNet to some extent by pruning. Pruning strategies were applied in CBNet when *K* = 2. As shown in Figure 5, *p_i_* represents pruning *i* stages from the lead backbone; two backbones share the first *i* stages. The *i+*1th stage transformation is as shown in Equation (13):(13)$$x_{2}^{i+1}=F_{C}^{i+1}(Cat(Cat(x_{1}),x_{1}^{i})),\;0\leq i\leq 6$$
where $Cat(x_{1})$ represents the connection from the auxiliary backbone to the lead backbone. When *i* = 0, the network is not pruned; when *i* = 6, the network is a single backbone network. According to Section 4.3, enhancement results from FCC and DHLC achieved excellent scores, and both FCC and DHLC variants were applied for subsequent studies.

We conducted analyses of underwater image enhancement from networks with different pruning strategies, as shown in Figure 6, details of analyses from different pruning strategies as shown in Appendix A. According to the SSIM evaluation in Figure 6a, FCC obtained better SSIM scores than DHLC. Both FCC and DHLC obtained the best enhancement result when pruning two stages. Moreover, FCC(*p*_2_) achieved SSIM_gray_ = 0.9409 and SSIM_RGB_ = 0.9119, which is a breakthrough in full-reference underwater image enhancement. According to the MSE evaluation in Figure 6b, the best MSE scores for FCC and DHLC were also obtained when pruning two stages, with MSE scores for FCC being slightly lower than those for DHLC. According to Figure 6c, the best PSNR scores for DHLC were obtained in *p*_2_. On the contrary, the best PSNR scores for FCC were obtained in *p*_3_ and *p*_4,_ respectively. According to a runtime evaluation in Figure 6d, the runtime of FCC had an inverse relationship with pruning stages, with runtime decreasing smoothly with an increase in pruning stages. The runtime of FCC remained in the range of 1.4 × 10^−3^~2.6 × 10^−3^ s. The runtime of DHLC from *p*_0_ to *p*_3_ was excessively long, and there was a sudden decline in runtime between *p*_3_ and *p*_4_. In conclusion, FCC(*p*_2_) obtained the best underwater image enhancement results, and DHLC(*p*_2_)-enhanced results obtained the best PSNR scores. These architectures were applied in subsequent studies.

### 4.5. Auxiliary Loss Analysis

Considering the special architecture of AHLC and DHLC, output information of the lead backbone’s last stage was directly transmitted to Output Conv Block. As a result, the input information of Output Conv Block had the same channel number as the output information of the auxiliary backbones. Thus, auxiliary losses could be introduced in the above variants, and information from the auxiliary backbones was sent to the Output Conv Block, generating an auxiliary image. The auxiliary losses were computed as shown in Equation (14):

(14)$$L(I_{aux}^{k},I_{R})=\omega_{l_{1}}L_{l_{1}}(I_{aux}^{k},I_{R})+\omega_{\mathrm{SSIM}}L_{\mathrm{SSIM}}(I_{aux}^{k},I_{R})$$
where $I_{aux}^{k}$ represents the auxiliary image from the *k*th backbone, *k* = 1, 2, …, *k* − 1. The total loss was constructed as a weighted sum of the main loss as well as auxiliary losses, as shown in Equation (15):(15)$$L_{total}=L(I_{P},I_{R})+\sum_{k=1}^{K-1}\lambda_{k}L(I_{aux}^{k},I_{R})$$
where *λ_k_* is weight for auxiliary loss from $I_{aux}^{k}$.

We introduced auxiliary losses in AHLC, DHLC as well as DHLC(*p_2_*) according to Section 4.3 and Section 4.4. The architecture is shown in Figure 7. Moreover, we put two auxiliary strategies into the experiment; one is the full-auxiliary strategy (written down as *f-a*), where loss weight is selected as a constant that *λ*_1_ = 0.3, and the other is the semi-auxiliary strategy (written down as *s-a*) where during first 50 epochs, *λ*_1_ = 0.3 and *λ*_1_ = 0 later on.

We evaluated semi-auxiliary and full-auxiliary strategies underwater image enhancement results, then compared them with enhancement results without auxiliary losses. Evaluations of underwater image enhancement results are shown in Figure 8, Figure 9 and Figure 10, details as shown in Appendix A.

According to SSIM evaluation (Figure 8) and MSE evaluation (Figure 9), AHLC obtained the best scores in the semi-auxiliary strategy (MSE_RGB_ = 161.22), which is also a breakthrough in MSE evaluation. For DHLC, the full-auxiliary strategy and the method without auxiliary losses obtained better results than the semi-auxiliary strategy. DHLC(*p_2_*) obtained the best scores without auxiliary losses. However, according to Figure 10, auxiliary strategies did not perform well in RSNR. As auxiliary losses are abandoned during network testing, the introduction of auxiliary losses does not affect the network runtime.

### 4.6. Summary

According to the comprehensive analyses above, various network architectures and optimization strategies of CBNet have achieved significant effects in underwater image enhancement. Among them, FCC with two pruned stages (FCC(*p*_2_)), DHLC with two pruned stages (DHLC(*p*_2_)), and AHLC with the semi-auxiliary strategy (AHLC(*s-a*)) obtained excellent image enhancement results. We conducted a comparison among the three networks, as shown in Table 3, where FCC(*p*_2_) obtained the best scores in SSIM evaluations, DHLC(*p*_2_) obtained the best scores in PSNR evaluations, and AHLC(*s-a*) obtained the best scores in MSE_RGB_. Considering FCC(*p*_2_) can obtain effective underwater image enhancement in the shortest time, hence meeting the demand of real-time performance, FCC(*p*_2_) was selected as the optimal underwater image-enhancing composite backbone network (OECBNet).

## 5. Cross-Sectional Research

To verify the performance of our proposed OECBNet, we compared OECBNet with several state-of-the-art CNN-based methods, including UWCNN [9], Water-Net [19], UIEC2-Net [21], IW-Net [23], and MC-CBNet [22]. Besides the 90 pairs of images from the UIEB reference subset, 60 images from the UIEB challenge subset were applied to the analysis. Both full-reference comparison and non-reference comparison were applied in the cross-sectional study. We applied two indicators in the non-reference comparison, one being underwater color image quality evaluation (*UCIQE*) [40], which consists of color density *σ_c_*, saturation *μ_s,_* and contrast *con_l,_* as shown in Equation (16),
(16)$$UCIQE=0.4680\sigma_{c}+0.2745con_{l}+0.2576\mu_{s},$$
and the other being underwater image quality measure (*UIQM*) [41], which is a combination of three sub-indicators: underwater image colorfulness measurement (*UICM*), underwater image sharpness measurement (*UISM*), and underwater image contrast measurement (*UIConM*). The expression is shown in Equation (17):(17)UIQM=0.0282UICM+0.2953UISM+3.5753UIConM

### 5.1. Reference Subset Evaluation

First, we tested our network as well as state-of-the-art CNN-based methods in the reference subset, and some underwater image enhancement results are shown in Figure 11.

As they are affected by natural illumination and differences in water quality, underwater images bear different features. In shallow water, the color features of images are reserved greatly due to the relatively high illumination. As the light gradually spreads into the deep underwater, the red, green, and blue spectra disappear in succession; hence, many underwater images tend to have cool tones, such as green and blue. On the other hand, images from the water–sediment interphase tend to be yellowish-brown, while images from low-illumination areas appear dim. According to the subjective comparison shown in Figure 11, it is visible that our proposed OECBNet obtained images the closest to reference images from high illumination, low illumination, and greenish or blueish underwater images and obtained clearer images from yellowish underwater images even than reference images. This indicated that OECBNet effectively eliminated the blue and green color bias in underwater images, reduced the effect of sediment on underwater images, and accurately expressed the color of objects and the environment. In addition, our proposed method retained detailed features of the underwater images and enhanced them to appear brighter and clearer.

A full-reference comparison of OECBNet as well as state-of-the-art CNN-based methods is shown in Table 4. OECBNet obtained the best scores in SSIM and MSE evaluations but did not achieve the best PSNR scores, indicating that images obtained from OECBNet still bear great deviations in some pixels.

A non-reference comparison of OECBNet and the state-of-the-art CNN-based methods is shown in Table 5. OECBNet obtained the best scores in *UCIQE*, *UIQM*, *UICM*, and *UISM* and had better scores than reference images in *UCIQE*, *UIQM*, *UISM*, and *UIConM*, which verified the excellent performance of our proposed network.

The runtime analysis is shown in Table 6. Our proposed network had an obvious real-time advantage compared with most underwater image enhancement networks and achieved underwater image enhancement much swifter than state-of-the-art CNN-based methods, except for UWCNN. It can be seen that the optimization of the architecture has achieved an obvious effect on time complexity and effectively solved the problem of the runaway growth of the time complexity caused by the complex network.

### 5.2. Challenging Subset Evaluation

To further verify the excellent image enhancement capability of our network, we tested our network against the state-of-the-art CNN-based methods using a challenging subset from the UIEB dataset. The results are shown in Figure 12.

Our proposed method eliminated color cast and obtained more realistic images than other strategies. For underwater images from low-visibility environments, OECBNet clearly restored the underwater scene. According to the challenge subset, the texture structures of the images are relatively simple, and the visibility is relatively low. It is relatively harder to see differences among the enhanced images from different algorithms on the macro level. Although there were no significant differences among IW-Net (e), MC-CBNet (f), and OECBNet (g) enhanced results, our proposed OECBNet achieved better color control in detail, which can be clearly represented by quantitative evaluation.

Due to the lack of reference images in the challenging subset, we applied non-reference indicators to quantitatively evaluate the enhanced underwater images. The comparison of OECBNet with state-of-the-art CNN-based methods is shown in Table 7. OECBNet-enhanced images obtained the best scores in *UCIQE*, *UIQM*, *UICM*, and *UISM*, hence achieving the best underwater image enhancement results.

## 6. Conclusions

In this paper, we applied a composite backbone network (CBNet) in underwater image enhancement by first constructing the global architecture of the proposed CBNet as well as backbone details, then conducting comprehensive research on the architecture of CBNet. Since the backbone is an important part of CBNet, we analyzed the impact of backbone number for CBNet-based underwater image enhancement, then compared the performances of different connection strategies in CBNet, which includes same-level composition (SLC), adjacent higher-level composition (AHLC), adjacent lower-level composition (ALLC), dense same-level composition (DSLC), dense higher-level composition (DHLC), and full-connected composition (FCC). Due to the suitable performance of DHLC and FCC, both of them were selected for pruning strategy analysis. In addition, the auxiliary loss was also applied in CBNet research. To summarize, we selected a network with the best enhancement results as the optimal underwater image-enhancing composite backbone network (OECBNet) for subsequent research. According to the cross-sectional research conducted, the images obtained by our proposed network accurately expressed the colors of objects as well as the colors of corresponding environments while retaining fine feature details. According to quantitative analyses, our proposed network obtained better scores than state-of-the-art CNN-based methods in both full-reference indications and non-reference indications. In addition, our proposed network also performed well in real-time image enhancement and achieved a record 2.3 × 10^−3^ s per image. We postulate that CBNet, with more than three backbones and multiple connection strategies, might achieve better performance in underwater image enhancement and recommend further research into these architectures. We expect that our proposed underwater image enhancement network will be applied in ROV for taking underwater videos and underwater detection.

## Figures and Tables

**Figure 1 biomimetics-08-00275-f001:**
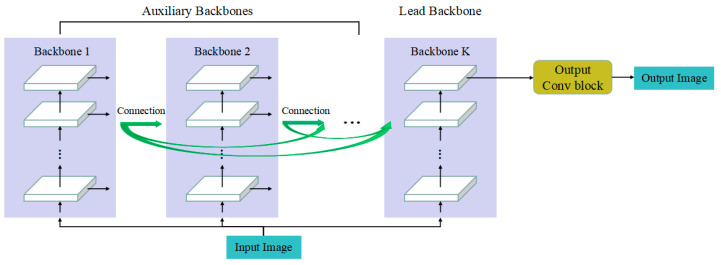
General architecture of CBNet for image enhancement.

**Figure 2 biomimetics-08-00275-f002:**
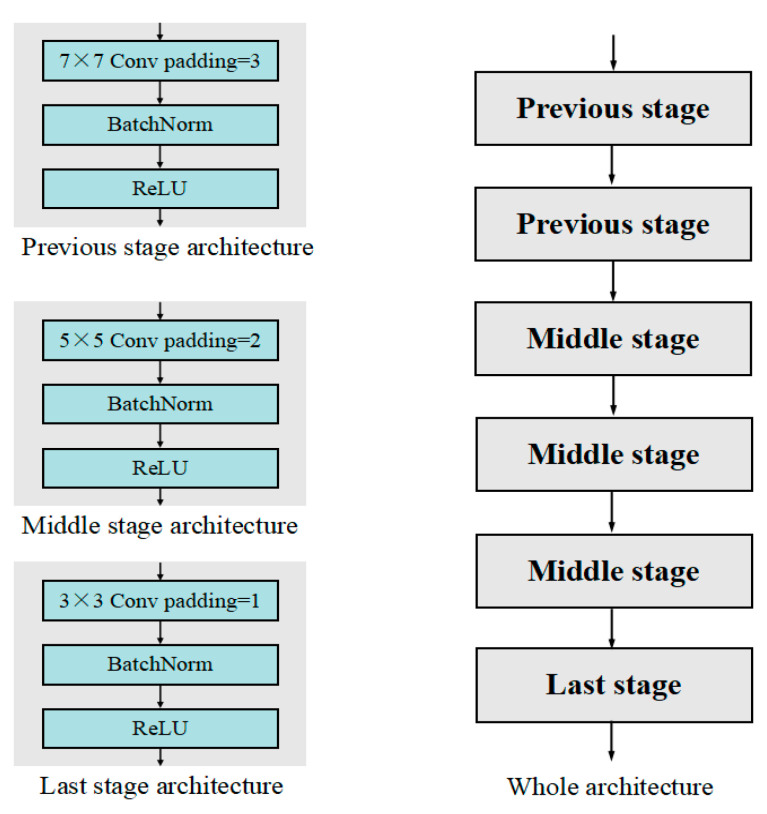
Architecture details of backbones.

**Figure 3 biomimetics-08-00275-f003:**
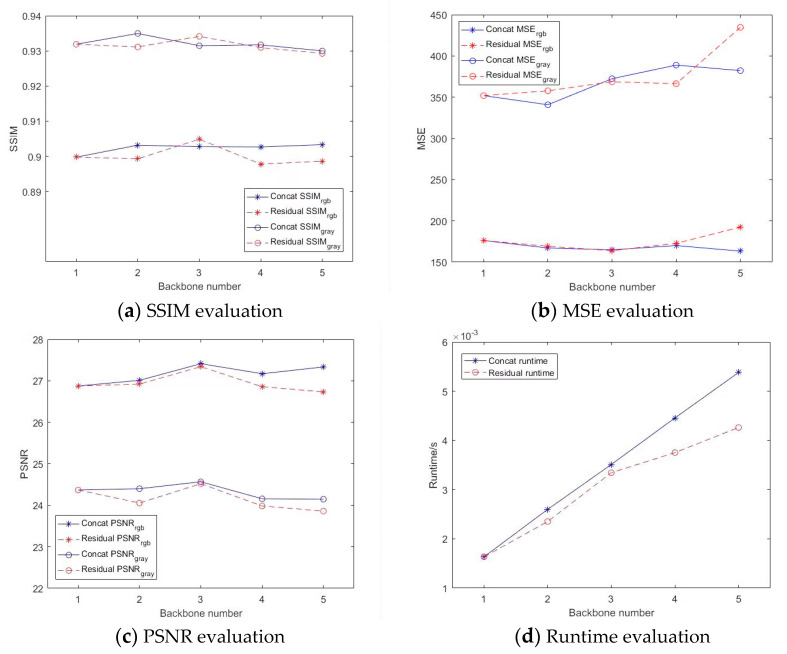
Analyses from different backbone numbers.

**Figure 4 biomimetics-08-00275-f004:**
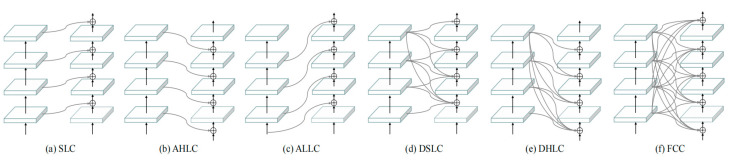
Architecture of different composite strategies for CBNet.

**Figure 5 biomimetics-08-00275-f005:**
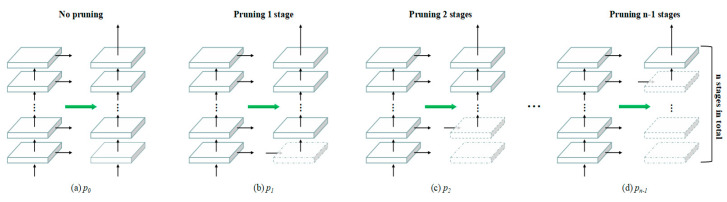
Architecture of pruning strategies for CBNet.

**Figure 6 biomimetics-08-00275-f006:**
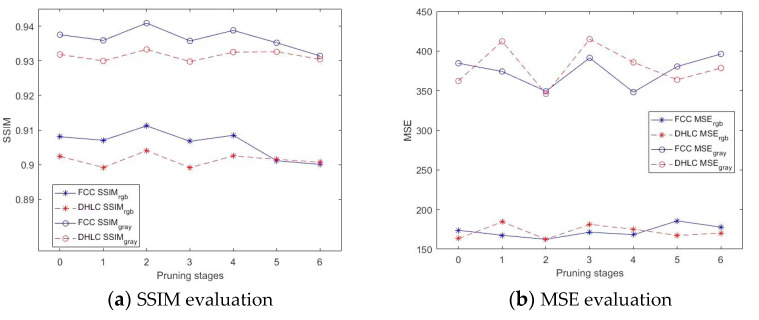
Analyses from pruning strategies.

**Figure 7 biomimetics-08-00275-f007:**
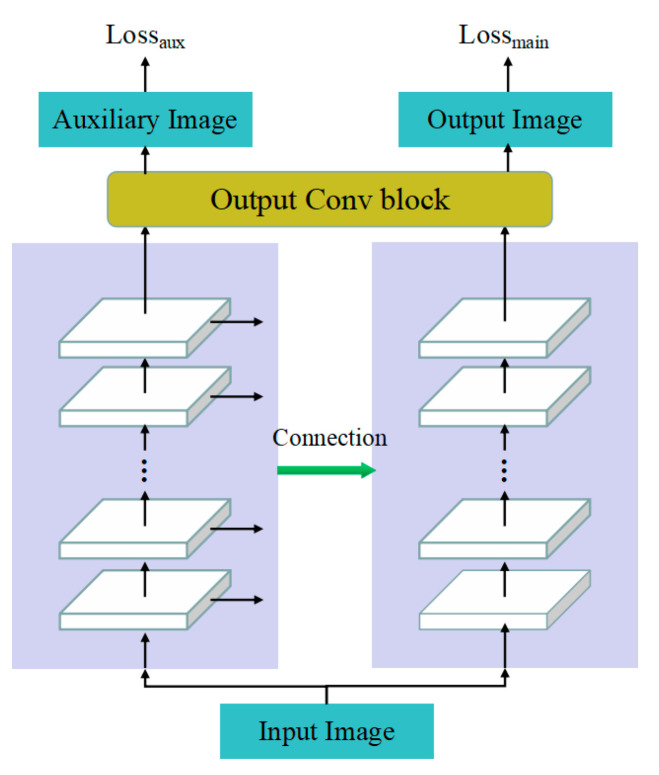
Architecture of CBNet (*K* = 2) with auxiliary loss.

**Figure 8 biomimetics-08-00275-f008:**
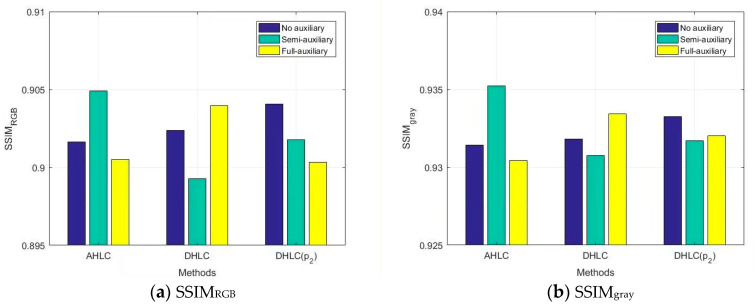
SSIM evaluation on auxiliary loss analysis.

**Figure 9 biomimetics-08-00275-f009:**
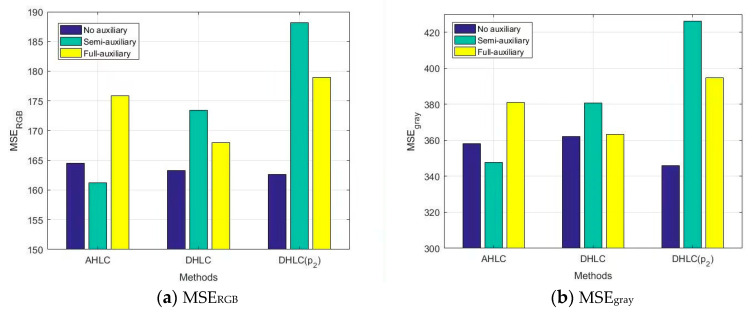
MSE evaluation on auxiliary loss analysis.

**Figure 10 biomimetics-08-00275-f010:**
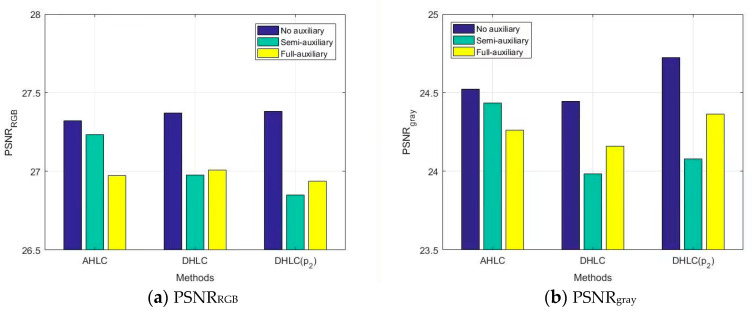
PSNR evaluation on auxiliary loss analysis.

**Figure 11 biomimetics-08-00275-f011:**
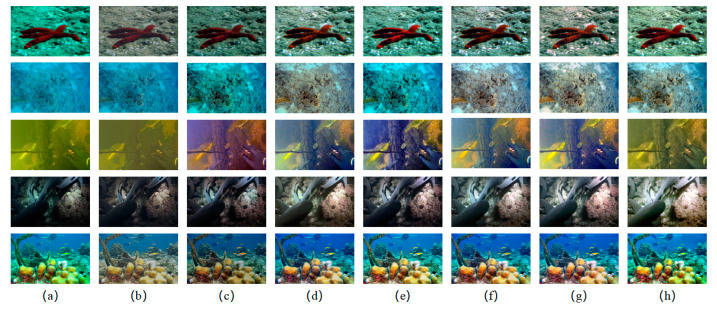
Subjective comparison of underwater image enhancement results using a reference subset. (**a**) Raw images. (**b**) Results from UWCNN. (**c**) Results from Water-Net. (**d**) Results from UIEC2-Net. (**e**) Results from IW-Net. (**f**) Results from MC-CBNet. (**g**) Results from the proposed OECBNet. (**h**) Reference images.

**Figure 12 biomimetics-08-00275-f012:**
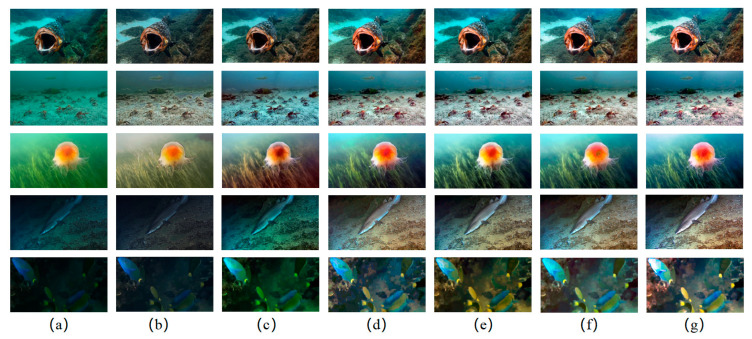
Subjective comparison of underwater image enhancement results using a challenging subset. (**a**) Raw images. (**b**) Results from UWCNN. (**c**) Results from Water-Net. (**d**) Results from UIEC2-Net. (**e**) Results from IW-Net. (**f**) Results from MC-CBNet. (**g**) Results from the proposed OECBNet.

**Table 1 biomimetics-08-00275-t001:** Enhancement results evaluation for different connection strategies.

Method	SSIM_RGB_	SSIM_gray_	MSE_RGB_	MSE_gray_	PSNR_RGB_	PSNR_gray_
SLC	0.9025	0.9323	172.95	390.66	27.224	24.374
AHLC	0.9017	0.9314	164.52	358.08	27.322	24.523
ALLC	0.9045	0.9344	170.36	369.54	26.983	24.107
DSLC	0.9059	0.9350	164.38	371.45	27.201	24.201
DHLC	0.9024	0.9318	163.32	362.24	27.370	24.445
FCC	0.9080	0.9375	173.81	384.59	27.026	24.047

**Table 2 biomimetics-08-00275-t002:** Runtime comparison for networks with different connection strategies.

Method	Average Runtime/s
SLC	0.00261
AHLC	0.04381
ALLC	0.00255
DSLC	0.13251
DHLC	0.13596
FCC	0.00259

**Table 3 biomimetics-08-00275-t003:** Comparison of evaluation results for three excellent networks.

Method	SSIM_RGB_	SSIM_gray_	MSE_RGB_	MSE_gray_	PSNR_RGB_	PSNR_gray_	Runtime
FCC(*p*_2_)	0.911	0.941	162.4	349.3	27.09	24.32	0.0023
DHLC(*p*_2_)	0.904	0.933	162.6	345.9	27.38	24.72	0.0514
AHLC(*s-a*)	0.905	0.935	161.2	347.7	27.23	24.44	0.0438

**Table 4 biomimetics-08-00275-t004:** Full-reference comparison of enhancement results in reference subset.

Method	SSIM_RGB_	SSIM_gray_	MSE_RGB_	MSE_gray_	PSNR_RGB_	PSNR_gray_
UWCNN	0.8443	0.8799	496.32	1191.6	22.550	19.470
Water-Net	0.8840	0.9188	380.14	874.80	23.738	20.946
UIEC^2^Net	0.9034	0.9336	168.35	368.49	27.039	24.238
IW-Net	0.9062	0.9348	223.10	473.09	27.098	24.496
MC-CBNet	0.9048	0.9349	168.11	385.16	27.375	24.553
OECBNet	0.9112	0.9409	162.40	349.29	27.091	24.322

**Table 5 biomimetics-08-00275-t005:** Non-reference comparison of enhancement results in reference subset.

Method	*UCIQE*	*UIQM*	*UICM*	*UISM*	*UIConM*
Raw	0.4525	1.9821	4.9778	3.5095	0.2252
UWCNN	0.4416	2.5893	4.0856	4.6135	0.3109
Water-Net	0.5341	2.5043	5.4714	4.5885	0.2783
UIEC^2^Net	0.5288	2.6618	6.5597	4.8831	0.2895
IW-Net	0.5559	2.5874	6.8383	4.7999	0.2733
MC-CBNet	0.5520	2.6580	6.9278	4.8821	0.2856
OECBNet	0.5612	2.7178	7.1059	5.0357	0.2882
Reference	**0.5595**	**2.5024**	**7.5522**	**4.6337**	**0.2576**

**Table 6 biomimetics-08-00275-t006:** Runtime analysis of enhancement results.

Method	Average Runtime/s
UWCNN	0.00143
Water-Net	0.23477
UIEC^2^Net	0.11010
IW-Net	0.31910
MC-CBNet	0.37713
OECBNet	0.00230

**Table 7 biomimetics-08-00275-t007:** Quantitative evaluation of enhancement results in the challenging subset.

Method	*UCIQE*	*UIQM*	*UICM*	*UISM*	*UIConM*
Raw	0.3907	1.6933	3.1112	2.4060	0.2504
UWCNN	0.3877	2.1985	2.5470	3.3318	0.3196
Water-Net	0.5132	2.1455	4.4568	3.2896	0.2932
UIEC^2^Net	0.5056	2.3049	4.7146	3.7595	0.2970
IW-Net	0.5369	2.2320	4.7800	3.4878	0.2985
MC-CBNet	0.5302	2.2942	5.0663	3.7356	0.2932
OECBNet	0.5503	2.3240	5.3689	3.7981	0.2940

## Data Availability

The data presented in this study are available upon request from the corresponding author. The data are not publicly available due to it being only available to teams interested in collaboration.

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
