# Peer review of "A Novel Underwater Image Enhancement Using Optimal Composite Backbone Network"

_biomimetics, 2023, doi:10.3390/biomimetics8030275_

Round 1
Reviewer 1 Report
1. Please provide a detailed analysis of the differences between the method proposed in this paper and the recently proposed methods in the underwater image enhancement task, such as: SGUIE-Net: Semantic Attention Guided Underwater Image Enhancement with Multi-Scale Perceptionï¼›Multi-view underwater image enhancement method via embedded fusion mechanismï¼›Underwater image enhancement method by multi-interval histogram equalizationï¼›Cross-view enhancement network for underwater imagesï¼›Auto Color Correction of Underwater Images Utilizing Depth Informationï¼›Underwater Image Enhancement Quality Evaluation: Benchmark Database and Objective Metric
2. It's hard to see the difference between (c)(f)(g) in Figure 12. Please analyze the reasons for the poor image quality caused by each method.
3. The manuscript states that one of the goals of the proposed method is to shorten the running time. It would be helpful for the authors to include a comparison of running times between their proposed method and other state-of-the-art methods, demonstrating the real-time performance advantage of OECBNet. The author should give a detailed analysis of the reasons for the high time complexity of other methods and the low time complexity of this method.
4. Some of the references cited in the manuscript are not properly formatted. Please ensure that all references follow the journal's citation style.
5. The manuscript uses several terms and acronyms that may be unfamiliar to readers. Please provide definitions or explanations for these terms when they are first introduced in the text.
Moderate editing of English language
Reviewer 2 Report
This paper proposed an improved CBNet variant called OECBNet to enhance the underwater images. Detailed experiments were conducted on the UIEB dataset and ablation studies of the architecture including backbone numbers, connection, pruning strategy, and loss functions are done. However, there are still some points not clear in this paper.
Point 1: As shown in Fig.2 and described in section 3.2, 2 previous stages, 3 middle stages and 1 last stage compose the whole backbone. Is there an ablation study about how many blocks in the backbone are thought to be optimal? Or is the architecture of one single backbone chosen empirically? Intuitively, the “middle stage” should be one big block composed of several tiny blocks. The authors may want to add more explanations about this section.
Point 2: “According to SSIM evaluations in Fig.3(a), networks with concat fusion obtained 232 the best SSIM score when K = 3, and networks with residual fusion obtained the best SSIM 233 score when K = 2.” (Starts from line 232) For the concat fusion group specifically, neither SSIM_RGB nor SSIM_gray has higher SSIM value than the residual ones, and the highest scores for both metrics are obtained when K=2 and K=5 instead of K=3 according to Fig.3(a). The same mistake appears in the residual group. The authors should explain more about this issue.
Point 3: Authors may want to highlight the best performance in Tables for each method and metric to make the conclusion more clear.
Point 4: There are still some small mistakes in the paper, the authors should read through the paper more carefully and correct the issues. For example, the pi (line 295) may be referring to a symbol p_i.
None
Author Response
Thank you for your proposed points, your advice are very helpful to our article.

Reviewer 3 Report
This article contains a variety of information related to the topics of image processing and neural networks. Moreover, the original Optimal Underwater Image Enhancing Composite Backbone Network architecture is proposed. It is also worth noting that the work has remarkable clarity compared to existing methods. The article is enriched with a multitude of diagrams and charts, which simplify the understanding of the material presented. However, the reviewer has a few comments and questions:
1. At the end of the "Introduction" section, there is no short description of each paper’s section.
2. Line 63 lacks a description about image dimensions, whereas real-time processing is highly time-dependent on information arrays. Lines 181-182 give a brief description of the training image dimensions, but it should be mentioned earlier.
3. The results should be described in an impersonal manner.
4. There is no justification as to why neural networks are better than basic processing algorithms for improving image quality. What exactly is "better"?
5. The training time of the neural network on the presented hardware should be given.
6. MSE metric does not give an accurate measure of neural network quality, perhaps other metrics should be considered (e.g. FPE) or text interpretation of MSE results should be introduced.
The article requires minor revision in terms of declared remarks with the same results obtained.
English may increase the number of checks in terms of small adjustments to sentences.
Reviewer 4 Report
The paper is interesting. However, the paper should be reorganized. For example, there is no Results section. Results are mixed with proposed and with known methods and explanations. You should separate proposed modification into separate section. Also, Results should be separated section. Previous works belong to different introductory or methodology section.
Figs 11 and 12 should be described in text. You should tell the reader what is visible and why.
Language is fine, but it could be improved. Although understandable, some sentences are too long, and could be divided into smaller.
Round 2
Reviewer 4 Report
In future, please provide detailed list of changes and answers to comments and separate highlighted revised paper with different colors for different reviewers.
My comments in the previous round wasn't answered or debuted or denied with arguments.
Author Response
We highlighted detailed descriptions of Fig. 11 and Fig. 12, and told the reader what is/isn't visible and why.